# scJEPA: Dual-Space Self-Supervised Learning for Single-Cell Representations

**Yuyao Zhai, Weixu Wang, Till Richter & Fabian J Theis** [*]
Institute of Computational Biology
Helmholtz Munich
Germany
{yuyao.zhai, weixu.wang, till.richter, fabian.theis}@helmholtz-munich.de

## Abstract

Learning robust representations from large-scale single-cell transcriptomic data is essential for understanding cellular heterogeneity, yet most self-supervised approaches operate in only one of two spaces. Reconstruction-based methods effectively denoise gene expression but impose no constraints on the embedding space, while contrastive methods organize embeddings but do not explicitly denoise inputs. Here, we introduce scJEPA, a dual-space self-supervised framework: denoising reconstruction captures global structure in expression space, while cross-view latent prediction organizes the embedding space by enforcing that masked views share consistent representations. Thereby it retains only predictable biological signals while discarding view-specific noise such as dropout and batch effects. Crucially, each scJEPA objective operates in a distinct space with a specialized role: denoising guarantees information preservation, while latent prediction determines which information is retained. We systematically evaluate scJEPA against reconstruction-based and contrastive methods under large-scale pretraining settings. Across diverse tasks, including zero-shot cell-type retrieval, classification on held-out datasets, cross-batch integration, and transfer to spatial transcriptomics, scJEPA consistently outperforms single-space objectives. Our results demonstrate that jointly learning both in data and embedding spaces provides representations that better capture cellular properties.

## 1 Introduction

The rapid accumulation of single-cell transcriptomic data has created unprecedented opportunities to understand cellular heterogeneity at single-cell resolution. With datasets now routinely containing millions of cells, the field faces a critical challenge (Bunne et al., 2024): how can we learn representations capturing biological structure from such high-dimensional, sparse, and noisy data?

Self-supervised learning (SSL) offers a compelling solution by leveraging the data itself to define learning objectives without requiring massive annotations (Chen et al., 2020). In single-cell genomics, existing approaches fall into two main categories. **Reconstruction-based methods** (Eraslan et al., 2019) learn to compress and reconstruct gene expression profiles through autoencoder frameworks, with masked autoencoders representing a prominent variant that reconstructs randomly masked genes from context. **Contrastive and non-contrastive methods** (Bahrami et al., 2025) learn by organizing the embedding space: contrastive approaches pull together positive pairs (e.g., augmented views of the same cell) while pushing apart negative pairs, whereas non-contrastive approaches such as self-distillation avoid explicit negative samples but often require architectural asymmetries like stop-gradients or momentum encoders. A recent systematic study (Richter et al., 2025) demonstrated that both paradigms, when properly scaled, yield substantial improvements on downstream tasks such as cell-type classification and batch integration.

Despite their success, these paradigms have complementary limitations rooted in where their objectives operate. Reconstruction-based methods effectively denoise gene expression by learning to recover true signals from corrupted observations (Eraslan et al., 2019), capturing global data structure.

---

[*]Corresponding Author

However, their loss functions act solely in the expression space, imposing no explicit constraints on the embedding space, which may therefore lack the beneficial geometric organization for downstream tasks (Van Assel et al., 2025). Contrastive methods directly structure the embedding space, but they do not explicitly denoise the input; moreover, their reliance on augmentation strategies raises concerns about biological validity (Richter et al., 2025). Neither paradigm simultaneously addresses both spaces.

The Joint-Embedding Predictive Architecture (JEPA) (Assran et al., 2023) offers a principled alternative for embedding space organization: by predicting latent representations across views rather than reconstructing raw observations, JEPA captures only information that is shared and predictable while discarding view-specific noise. This property is particularly well suited for single-cell data, where view-specific noise such as technical dropout, batch effects, and measurement stochasticity is pervasive. Predictability across views therefore acts as a natural filter that preferentially preserves biologically meaningful signal while suppressing noise. Crucially, theoretical analysis (Balestriero & LeCun, 2025) proves that combining such predictive objectives with a drawn isotropic Gaussian Regularization (SIGReg) prevents representation collapse and produces optimal embeddings for downstream tasks. However, JEPA operates entirely in embedded space and does not explicitly denoise input data. This observation motivates our approach: **integrating autoencoder-based denoising in expression space with JEPA-style regularization in embedding space**. The former captures global structure through denoising, while the latter ensures well-organized embeddings for downstream tasks.

In this work, we introduce scJEPA, a dual-space self-supervised framework (referring to joint optimization in expression and embedding spaces, not the mathematical dual vector space) for single-cell gene expression data. Unlike generic multi-loss ensembles, each objective in scJEPA operates in a distinct space with a specialized role: denoising reconstruction in expression space guarantees that sufficient biological information is preserved, while cross-view latent prediction in embedding space determines which information is retained by enforcing consistency across masked views. To prevent collapse, we employ SIGReg (Balestriero & LeCun, 2025), which provides stronger theoretical guarantees than simpler covariance-based penalties by enforcing full distributional matching rather than just second-order statistics.

We systematically evaluate scJEPA under large-scale pretraining settings. Our experiments demonstrate superior performance on cell-type classification in both zero-shot and fine-tuned settings across held-out datasets spanning diverse biological contexts. Beyond classification, scJEPA produces representations that better preserve biological variation during cross-batch integration and transfer effectively to spatial transcriptomics, suggesting that the combined objective captures fundamental cellular properties generalizing across conditions and modalities.

These results establish dual-space self-supervised learning as a powerful paradigm for single-cell genomics. Concurrent with our work, JEPA-style approaches such as CellJEPA (ElSheikh et al., 2026) and GeneJEPA (Litman et al., 2025) have emerged, adopting transformer backbones with standard I-JEPA components (double encoders, EMA, stop-gradients). scJEPA takes a different path: building on LeJEPA (Balestriero & LeCun, 2025), we replace heuristic collapse prevention with principled distributional regularization, enabling a simpler single-encoder design with theoretical guarantees. While extending to transformers remains a natural future direction, our results demonstrate that the choice of training strategy can matter more than backbone complexity.

## 2 METHOD

We propose an SSL framework that integrates the theoretical foundations of JEPA with the representation learning capabilities of an autoencoder. Our method, termed **scJEPA**, employs masked view generation and isotropic Gaussian regularization to learn robust and transferable representations. The complete end-to-end training procedure for **scJEPA** is outlined in Algorithm 1.

### 2.1 PROBLEM SETUP AND NOTATION

Consider a batch of $N$ input samples denoted by $\mathbf{X} \in \mathbb{R}^{N \times D}$, where each sample has dimensionality $D$. The model is built upon an autoencoder comprising an encoder $f_{\boldsymbol{\theta}} : \mathbb{R}^D \to \mathbb{R}^K$ and a decoder $g_{\boldsymbol{\varphi}} : \mathbb{R}^K \to \mathbb{R}^D$, with $K$ representing the dimensionality of the latent embedding.

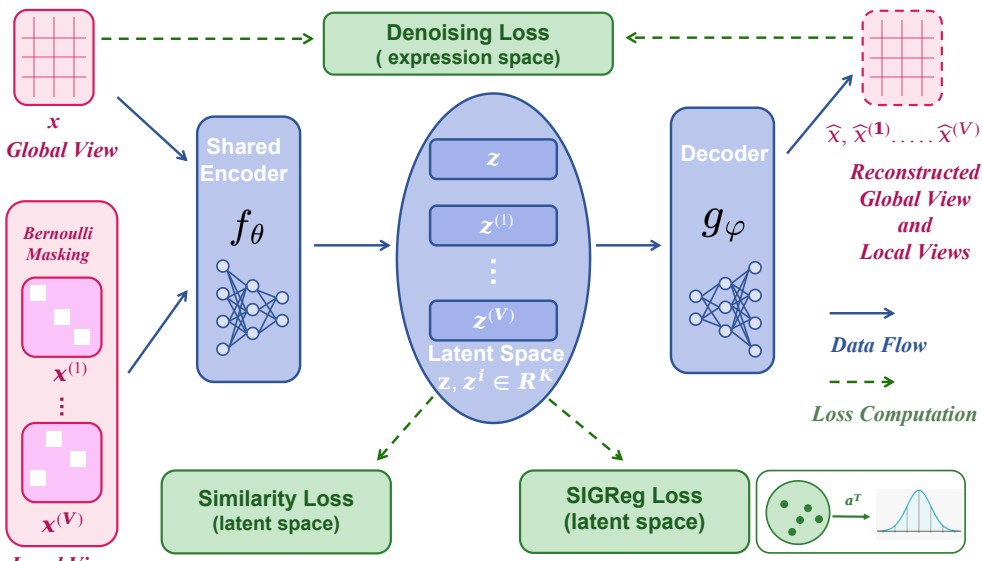

Figure 1: **Overview of scJEPA.** The encoder maps both the global view (unmasked) and multiple local views (masked) to a shared latent space. The training objective combines predictive similarity loss, SIGReg regularization, and denoising reconstruction.

## 2.2 MASKED VIEW GENERATION

To generate multiple views that retain the semantic content of the original input, we adopt a stochastic masking procedure. For an input sample $\mathbf{x} \in \mathbb{R}^D$, we produce $V$ masked views $\{\mathbf{x}^{(v)}\}_{v=1}^V$ by applying an element-wise Bernoulli mask:

$$\mathbf{x}^{(v)} = \mathbf{x} \odot \mathbf{m}^{(v)}, \quad \mathbf{m}_d^{(v)} \sim \text{Bernoulli}(1-p), \quad d = 1, \ldots, D, \tag{1}$$

where $\odot$ denotes the Hadamard product, $\mathbf{m}^{(v)} \in \{0,1\}^D$ is a binary mask for the $v$-th view, and $p \in (0,1)$ is the masking probability. We adopt random masking rather than structured alternatives (e.g., pathway-based or gene ontology-guided masking), as systematic evaluation in prior work demonstrated that random masking achieves competitive performance for single-cell SSL while being computationally simpler (Richter et al., 2025). In our experiments, we set $p = 0.5$ and generate $V = 8$ masked views per sample.

The original **unmasked** sample $\mathbf{x}$ is treated as the *global view*, while the masked variants $\{\mathbf{x}^{(v)}\}_{v=1}^V$ serve as *local views*. This design is analogous to the multi-crop strategy in vision transformers, adapted here for tabular gene expression data.

## 2.3 AUTOENCODER ARCHITECTURE

Both the global view and all local views are processed by the same shared autoencoder. The encoder maps each view to a latent representation:

$$\mathbf{z} = f_{\boldsymbol{\theta}}(\mathbf{x}), \quad \mathbf{z}^{(v)} = f_{\boldsymbol{\theta}}(\mathbf{x}^{(v)}), \quad v = 1, \ldots, V, \tag{2}$$

where $\mathbf{z}, \mathbf{z}^{(v)} \in \mathbb{R}^K$ are the corresponding latent embeddings.

The decoder subsequently reconstructs the input from the latent embeddings:

$$\hat{\mathbf{x}} = g_{\boldsymbol{\varphi}}(\mathbf{z}), \quad \hat{\mathbf{x}}^{(v)} = g_{\boldsymbol{\varphi}}(\mathbf{z}^{(v)}), \quad v = 1, \ldots, V. \tag{3}$$

## 2.4 TRAINING OBJECTIVE

The overall training loss comprises three components: a predictive similarity loss, an isotropic Gaussian regularization loss, and a denoising reconstruction loss.

### 2.4.1 PREDICTIVE SIMILARITY LOSS

Following the JEPA framework, we enforce that the embeddings of local masked views are predictable from the global unmasked view embedding. For a batch of $N$ samples, the similarity loss is defined as:

$$\mathcal{L}_{\text{sim}} = \frac{1}{NV} \sum_{n=1}^{N} \sum_{v=1}^{V} \left\| \mathbf{z}_n - \mathbf{z}_n^{(v)} \right\|_2^2, \tag{4}$$

where $\mathbf{z}_n$ is the global view embedding and $\mathbf{z}_n^{(v)}$ is the $v$-th local view embedding for sample $n$.

### 2.4.2 SKETCHED ISOTROPIC GAUSSIAN REGULARIZATION (SIGREG)

To prevent representation collapse and promote an embedding distribution favorable for downstream tasks, we regularize the latent embeddings to follow an isotropic Gaussian distribution $\mathcal{N}(\mathbf{0}, \mathbf{I}_K)$. A natural alternative would be simpler covariance-based penalties (e.g., decorrelation losses used in Barlow Twins (Zbontar et al., 2021)), which only enforce second-order statistics. However, matching covariance alone is insufficient: embeddings can satisfy covariance constraints while having arbitrary higher-order structure that harms downstream performance. SIGReg addresses this by enforcing full distributional matching through the Epps-Pulley characteristic function test, which captures all moments of the distribution (Balestriero & LeCun, 2025).

Specifically, given a set of $M$ randomly sampled unit-norm directions $\mathcal{A} = \{\mathbf{a}_1, \ldots, \mathbf{a}_M\}$ where $\mathbf{a}_m \sim \mathcal{U}(\mathcal{S}^{K-1})$, the SIGReg loss is:

$$\text{SIGReg}(\mathcal{A}, \{\mathbf{z}_n\}_{n=1}^{N}) = \frac{1}{M} \sum_{m=1}^{M} T_{\text{EP}} \left( \{\mathbf{a}_m^\top \mathbf{z}_n\}_{n=1}^{N} \right), \tag{5}$$

where $T_{\text{EP}}$ denotes the Epps-Pulley statistic:

$$T_{\text{EP}}(\{u_n\}_{n=1}^{N}) = N \int_{-\infty}^{\infty} \left| \hat{\phi}(t) - \phi_{\mathcal{N}}(t) \right|^2 w(t)\, dt. \tag{6}$$

Here, $\hat{\phi}(t) = \frac{1}{N} \sum_{n=1}^{N} e^{itu_n}$ is the empirical characteristic function, $\phi_{\mathcal{N}}(t) = e^{-t^2/2}$ is the characteristic function of the standard normal distribution, and $w(t) = e^{-t^2}$ is a Gaussian weighting function.

The total SIGReg loss aggregates the regularization across the global view and all local views:

$$\mathcal{L}_{\text{SIGReg}} = \frac{1}{V+1} \left[ \text{SIGReg}(\mathcal{A}, \{\mathbf{z}_n\}_{n=1}^{N}) + \sum_{v=1}^{V} \text{SIGReg}(\mathcal{A}, \{\mathbf{z}_n^{(v)}\}_{n=1}^{N}) \right]. \tag{7}$$

### 2.4.3 DENOISING LOSS

To ensure the learned representations retain sufficient information for reconstructing the original input, we incorporate a denoising reconstruction objective. The loss measures the mean squared error between the original inputs and their reconstructions:

$$\mathcal{L}_{\text{recon}} = \frac{1}{N} \sum_{n=1}^{N} \|\mathbf{x}_n - \hat{\mathbf{x}}_n\|_2^2 + \frac{1}{NV} \sum_{n=1}^{N} \sum_{v=1}^{V} \left\| \mathbf{x}_n - \hat{\mathbf{x}}_n^{(v)} \right\|_2^2. \tag{8}$$

For the masked local views, the reconstruction target is the original, unmasked input $\mathbf{x}_n$. This establishes a denoising autoencoder objective, encouraging the model to recover information that was deliberately omitted.

### 2.4.4 TOTAL LOSS

The final objective function is a weighted combination of the three loss terms:

$$\mathcal{L}_{\text{total}} = (1 - \lambda)\mathcal{L}_{\text{sim}} + \lambda \mathcal{L}_{\text{SIGReg}} + \gamma \mathcal{L}_{\text{recon}}, \tag{9}$$

where $\lambda \in (0, 1)$ balances the predictive similarity and distributional regularization, and $\gamma > 0$ controls the weight of the reconstruction loss.

## 2.5 THEORETICAL MOTIVATION

Our framework inherits theoretical guarantees from LeJEPA (Balestriero & LeCun, 2025) that are particularly relevant for single-cell data. First, enforcing an isotropic Gaussian embedding distribution provably minimizes downstream prediction risk for both linear and nonlinear probing tasks (Theorem 1 in (Balestriero & LeCun, 2025) ). This is especially valuable for single-cell analysis, where embeddings are routinely used for diverse downstream tasks without prior knowledge of the specific task (Lopez et al., 2018). Second, SIGReg provably prevents representation collapse without heuristics such as stop-gradients or momentum encoders. Collapse is a significant concern in single-cell settings where cell-type frequencies are highly imbalanced; rare populations can easily be absorbed into dominant clusters without explicit regularization (Xu et al., 2021). Third, the Epps-Pulley statistic yields bounded gradients (Theorem 4 in (Balestriero & LeCun, 2025)), ensuring stable optimization even with the large batch sizes required for million-scale pretraining.

The reconstruction objective complements these guarantees by ensuring that the latent space preserves sufficient information for input recovery. This is particularly relevant for single-cell data (Huang et al., 2018), where technical dropout and measurement noise are prevalent—the denoising objective encourages robustness to such corruptions while capturing global expression structure.

## 3 EXPERIMENTS

### 3.1 EXPERIMENTAL SETUP

**Datasets.** For pretraining, we use the scTab (Fischer et al., 2024) dataset from CELLxGENE census (Program et al., 2025), comprising 22.2 million human scRNA-seq profiles across 164 cell types, split by donor into 15.2M/3.5M/3.4M for training/validation/test. We evaluate on four task categories: (1) *In-distribution*: HLCA (Sikkema et al., 2023) (775K cells, 51 types), PBMCs (Yoshida et al., 2022) (78K cells, 30 types), and Tabula Sapiens (Consortium* et al., 2022) (336K cells, 161 types); (2) *Out-of-distribution*: seven held-out datasets including brain tissues (Siletti et al., 2023; Velmeshev et al., 2023; Ivanova et al., 2023; Jorstad et al., 2023), developing neocortex (Wang et al., 2025), and breast cancer (Janesick et al., 2023); (3) *Multi-batch integration*: three lung datasets with varying health conditions (Travaglini et al., 2020; Wang et al., 2020; Melms et al., 2021); (4) *Spatial transfer*: MERSCOPE human neocortex (Wang et al., 2025).

**Baselines and Evaluation.** We compare against two strong SSL baselines from Richter et al. (2025): masked autoencoder (MAE) and Barlow Twins (BT), evaluating both zero-shot (frozen encoder) and fine-tuned settings. We also include PCA and purely supervised training as reference points. For cell type prediction, we report $F1_{\text{Macro}}$ and $F1_{\text{Micro}}$. For data integration, we use scIB metrics (Luecken et al., 2022), which aggregate batch correction and biological conservation scores.

**Implementation.** Data undergo standard preprocessing (normalization to 10,000 counts, log1p). Our autoencoder uses hidden dimensions [512, 512, 256, 256, 64] with a 64-dim projection head. We train with AdamW (lr=0.001, weight decay=0.05) and StepLR scheduling. Pretraining takes $\sim$2 days on a single GPU with batch size 1024; fine-tuning requires $\sim$1 day.

### 3.2 RESULTS

#### 3.2.1 IN-DISTRIBUTION CELL TYPE ANNOTATION

Table 1: In-distribution cell type annotation.

| Method | scTab test set | | PBMC | | Tabula Sapiens | | HLCA | |
|---|---|---|---|---|---|---|---|---|
| | $F1_{\text{Macro}}$ | $F1_{\text{Micro}}$ | $F1_{\text{Macro}}$ | $F1_{\text{Micro}}$ | $F1_{\text{Macro}}$ | $F1_{\text{Micro}}$ | $F1_{\text{Macro}}$ | $F1_{\text{Micro}}$ |
| Supervised | 0.825 | 0.904 | 0.682 | 0.842 | 0.253 | 0.449 | 0.935 | 0.950 |
| Zero-Shot scMAE | 0.629 | 0.797 | 0.435 | 0.658 | 0.242 | 0.379 | 0.670 | 0.745 |
| Zero-Shot scJEPA | 0.726 | 0.859 | 0.653 | 0.798 | 0.341 | **0.499** | 0.851 | 0.899 |
| Finetune scMAE | 0.819 | 0.898 | 0.740 | **0.858** | 0.309 | 0.488 | 0.937 | 0.950 |
| Finetune scJEPA | **0.828** | **0.901** | **0.777** | 0.857 | **0.354** | 0.489 | **0.938** | **0.953** |

Table 1 presents results on datasets drawn from the same distribution as pretraining data. In the zero-shot setting, scJEPA substantially outperforms scMAE across all four datasets. On the scTab test set,

scJEPA achieves 0.726 $F1_{\text{Macro}}$ compared to 0.629 for scMAE—a relative improvement of 15.4%. The advantage is even more pronounced on HLCA (0.851 vs. 0.670, +27.0%) and PBMCs (0.653 vs. 0.435, +50.1%), demonstrating that the JEPA objective learns representations better aligned with cell type identity without any task-specific supervision.

After fine-tuning, scJEPA maintains competitive or superior performance. On PBMCs, fine-tuned scJEPA achieves the highest $F1_{\text{Macro}}$ (0.777), surpassing both scMAE (0.740) and the purely supervised baseline (0.682). On Tabula Sapiens—a challenging dataset with 161 cell types—scJEPA leads in zero-shot $F1_{\text{Macro}}$ (0.341 vs. 0.242) and remains competitive after fine-tuning. These results suggest that predicting in latent space encourages the model to capture cell-type-discriminative features that transfer effectively to classification tasks.

### 3.2.2 OUT-OF-DISTRIBUTION GENERALIZATION

Table 2: Out-of-distribution cell type annotation.

| Method | $\text{OOD}_{\text{HiT}}$ $F1_{\text{Macro}}$ | $\text{OOD}_{\text{nn}}$ $F1_{\text{Macro}}$ | $\text{OOD}_{\text{Circ Imm}}$ $F1_{\text{Macro}}$ | $\text{OOD}_{\text{Cort Dev}}$ $F1_{\text{Macro}}$ | $\text{OOD}_{\text{Great Apes}}$ $F1_{\text{Macro}}$ |
|---|---|---|---|---|---|
| Supervised | 0.905 | 0.449 | 0.106 | 0.039 | 0.126 |
| Zero-Shot scMAE | 0.898 | 0.253 | 0.090 | 0.042 | 0.101 |
| Zero-Shot scJEPA | 0.905 | 0.355 | 0.137 | 0.064 | 0.649 |
| Finetune scMAE | 0.899 | 0.527 | 0.118 | 0.050 | 0.294 |
| Finetune scJEPA | **0.999** | **0.597** | **0.161** | **0.066** | **0.445** |

Table 3: Out-of-distribution cell type annotation on additional held-out datasets.

| Method | Human neocortex | | Human breast cancer | |
|---|---|---|---|---|
| | $F1_{\text{Macro}}$ | $F1_{\text{Micro}}$ | $F1_{\text{Macro}}$ | $F1_{\text{Micro}}$ |
| Supervised | 0.796 | 0.860 | 0.888 | 0.910 |
| PCA | 0.807 | 0.870 | 0.888 | 0.899 |
| Zero-Shot scBT | 0.655 | 0.761 | 0.747 | 0.773 |
| Zero-Shot scMAE | 0.709 | 0.799 | 0.709 | 0.799 |
| Zero-Shot scJEPA | 0.675 | 0.783 | 0.775 | 0.773 |
| Finetune scBT | 0.750 | 0.827 | 0.868 | 0.874 |
| Finetune scMAE | 0.747 | 0.825 | 0.747 | 0.825 |
| Finetune scJEPA | **0.777** | **0.851** | **0.891** | **0.908** |

Tables 2 and 3 evaluate generalization to seven datasets completely held out from pretraining. Zero-shot scJEPA demonstrates strong robustness across diverse biological contexts. Most strikingly, on the Great Apes dataset, scJEPA achieves 0.649 $F1_{\text{Macro}}$, over 6× higher than scMAE and 5× higher than the supervised baseline. This remarkable transfer across species suggests that scJEPA captures evolutionarily conserved transcriptomic features rather than dataset-specific patterns.

After fine-tuning, scJEPA consistently outperforms alternatives. On HiT, scJEPA achieves near-perfect classification (0.999 $F1_{\text{Macro}}$). On the challenging circulating immune and cortical development datasets, scJEPA shows 36% and 32% relative improvements over scMAE, respectively.

On the additional OOD datasets (Table 3), fine-tuned scJEPA achieves the best performance on both human neocortex (0.777 $F1_{\text{Macro}}$) and breast cancer (0.891 $F1_{\text{Macro}}$). Notably, on breast cancer, scJEPA surpasses even the supervised baseline (0.888), demonstrating that SSL pretraining provides benefits beyond what task-specific supervision alone can achieve. The consistent improvements across brain, immune, and cancer contexts indicate that scJEPA learns broadly transferable biological representations.

### 3.2.3 MULTI-BATCH DATA INTEGRATION

Figure 2 evaluates data integration on three lung datasets exhibiting heterogeneous batch effects due to differences in study protocols and patient health status. We report scIB metrics across multiple evaluation runs to assess both performance and consistency.

scJEPA achieves the highest median scIB total score with relatively low variance, outperforming the dedicated integration method scVI and other SSL approaches. Decomposing the metric reveals the source of this advantage: scJEPA attains the strongest biological conservation, indicating superior

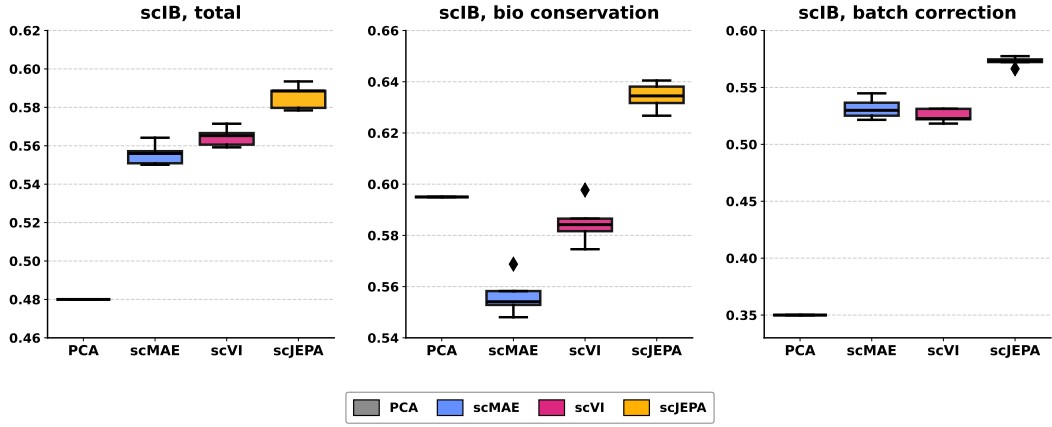

Figure 2: **Data integration on multi-source lung datasets.** scIB scores decomposed into biological conservation and batch correction components. Higher values indicate better performance.

preservation of cell-type identity during integration. For batch correction, scVI shows a slight edge, consistent with its design as a specialized integration method. However, this comes with a trade-off: aggressive batch correction can inadvertently remove genuine biological signal.

The results highlight a key strength of scJEPA: by regularizing embeddings to capture shared, predictable information across views, the model learns representations that emphasize intrinsic cellular properties over technical artifacts. This makes scJEPA particularly suitable for cross-study meta-analyses where preserving biological heterogeneity is paramount. The poor performance of PCA underscores the necessity of nonlinear representation learning for effective integration.

### 3.2.4 TRANSFER TO SPATIAL TRANSCRIPTOMICS

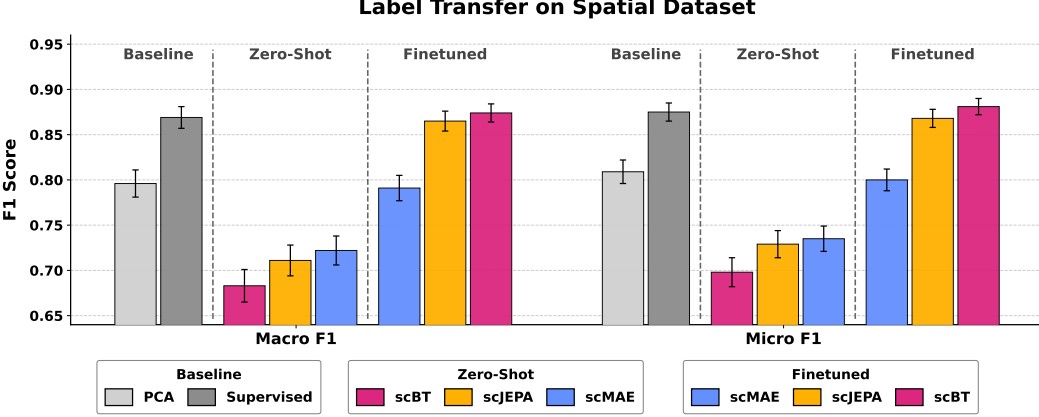

Figure 3: **Cross-platform transfer to spatial transcriptomics.** Cell type annotation performance on MERSCOPE human neocortex data, comparing Zero-Shot and Finetuned models.

Figure 3 assesses whether representations learned from dissociated scRNA-seq transfer to spatially-resolved measurements. This cross-platform evaluation is particularly challenging: spatial transcriptomics captures gene expression within intact tissue architecture, with distinct capture efficiencies and gene panels compared to droplet-based scRNA-seq.

After fine-tuning, scJEPA substantially outperforms scMAE and approaches the performance of both the supervised baseline and Barlow Twins. Notably, scJEPA achieves this competitive performance using a conceptually simpler objective that requires neither negative pairs (as in contrastive methods) nor momentum encoders (as in self-distillation approaches).

A striking pattern emerges when comparing zero-shot and fine-tuned performance: the improvement gap for scJEPA is notably smaller than that for scMAE, indicating that JEPA-pretrained representations are already well-suited for spatial data and require minimal adaptation. This property is

practically valuable in settings where labeled spatial transcriptomics data is scarce, as the pretrained representations provide a strong starting point even without extensive fine-tuning.

### 3.2.5 ABLATION STUDY

Table 4 examines the contribution of each component in scJEPA through systematic ablation on in-distribution datasets. We evaluate two variants: (1) *w/o reconstruction*, which removes the denoising objective $\mathcal{L}_{recon}$ and retains only the JEPA losses; (2) *w/o LeJEPA*, which removes both $\mathcal{L}_{sim}$ and $\mathcal{L}_{SIGReg}$, reducing the model to a standard denoising autoencoder.

Both components prove essential. Removing reconstruction causes substantial performance drops across all datasets, with HLCA $F1_{Macro}$ declining from 0.851 to 0.627, a 26% absolute decrease. This confirms that denoising in expression space captures global structure critical for cell-type discrimination. Removing LeJEPA also degrades performance, though less severely; for instance, PBMC $F1_{Macro}$ drops from 0.653 to 0.531. This indicates that embedding space regularization provides complementary benefits beyond what reconstruction alone achieves.

The ablation validates our dual-space design: neither denoising in expression space nor regularization in embedding space alone matches the full model. Their combination yields representations that capture both expression structure and well-organized embeddings suitable for downstream tasks.

Table 4: Ablation Study for the main modules in scJEPA. All methods use the pretrained models.

| Method | scTab test set | | PBMC | | Tabula Sapiens | | HLCA | |
|---|---|---|---|---|---|---|---|---|
| | $F1_{Macro}$ | $F1_{Micro}$ | $F1_{Macro}$ | $F1_{Micro}$ | $F1_{Macro}$ | $F1_{Micro}$ | $F1_{Macro}$ | $F1_{Micro}$ |
| scJEPA w/o reconstruction | 0.558 | 0.797 | 0.521 | 0.712 | 0.288 | 0.471 | 0.627 | 0.789 |
| scJEPA w/o LeJEPA | 0.673 | 0.813 | 0.531 | 0.695 | 0.299 | 0.465 | 0.771 | 0.821 |
| scJEPA | **0.726** | **0.859** | **0.653** | **0.798** | **0.341** | **0.499** | **0.851** | **0.899** |

## 4 CONCLUSION

We introduce scJEPA, a self-supervised framework that jointly optimizes in both expression and embedding spaces for single-cell transcriptomics. By combining denoising reconstruction with cross-view latent prediction regularized by SIGReg, scJEPA captures global expression structure while ensuring well-organized embeddings with theoretical guarantees against collapse. Systematic evaluation demonstrates consistent improvements over reconstruction-based and contrastive methods across zero-shot retrieval, cell-type annotation, batch integration, and cross-platform transfer to spatial transcriptomics.

Several directions remain for future work. First, our current implementation uses fully connected architectures; combining scJEPA's objective with transformers may better capture gene-gene dependencies and enable scaling to larger pretraining corpora. Second, extending the framework to multi-modal single-cell data (e.g., joint RNA and protein measurements) could yield unified representations across modalities. Third, our ablation reveals that pure LeJEPA without reconstruction underperforms on single-cell data, unlike its success in vision—this gap motivates that exploring domain-specific adaptations may reduce the reliance on explicit denoising for particular downstream tasks like trajectory inference or perturbation prediction.

### MEANINGFULNESS STATEMENT

We consider a meaningful representation of life to be one that captures the intrinsic biological identity of cells, such as their types, states, and functional programs, while remaining invariant to technical artifacts and measurement noise. Such representations should generalize across experimental conditions, datasets, and even species, reflecting evolutionarily conserved molecular principles rather than dataset-specific patterns.

Our work contributes to this vision by demonstrating that predicting in latent space, rather than reconstructing raw measurements, yields representations that better preserve biological signal. The striking cross-species transfer observed in our experiments suggests that scJEPA captures fundamental transcriptomic features shared across primates, moving closer to universal cellular representations.

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

## A   APPENDIX

### A.1   ALGORITHM SUMMARY

---

**Algorithm 1** End-to-end training procedure for scJEPA.

---

**Require:** Dataset $\mathcal{D}$, encoder $f_{\boldsymbol{\theta}}$, decoder $g_{\boldsymbol{\varphi}}$, masking probability $p$, number of views $V$, hyperparameters $\lambda$, $\gamma$

 1: **for** each mini-batch $\{\mathbf{x}_n\}_{n=1}^{N} \sim \mathcal{D}$ **do**
 2:      // Generate masked views
 3:      **for** $v = 1$ to $V$ **do**
 4:          Sample mask $\mathbf{m}^{(v)} \sim \text{Bernoulli}(1-p)^D$
 5:          $\mathbf{x}_n^{(v)} \leftarrow \mathbf{x}_n \odot \mathbf{m}^{(v)}$ for all $n$
 6:      **end for**
 7:      // Encode all views
 8:      $\mathbf{z}_n \leftarrow f_{\boldsymbol{\theta}}(\mathbf{x}_n)$ for all $n$
 9:      $\mathbf{z}_n^{(v)} \leftarrow f_{\boldsymbol{\theta}}(\mathbf{x}_n^{(v)})$ for all $n, v$
10:      // Decode all views
11:      $\hat{\mathbf{x}}_n \leftarrow g_{\boldsymbol{\varphi}}(\mathbf{z}_n)$ for all $n$
12:      $\hat{\mathbf{x}}_n^{(v)} \leftarrow g_{\boldsymbol{\varphi}}(\mathbf{z}_n^{(v)})$ for all $n, v$
13:      // Compute losses
14:      Compute $\mathcal{L}_{\text{sim}}, \mathcal{L}_{\text{SIGReg}}, \mathcal{L}_{\text{recon}}$
15:      $\mathcal{L}_{\text{total}} \leftarrow (1-\lambda)\mathcal{L}_{\text{sim}} + \lambda\mathcal{L}_{\text{SIGReg}} + \gamma\mathcal{L}_{\text{recon}}$
16:      Update $\boldsymbol{\theta}, \boldsymbol{\varphi}$ via gradient descent on $\mathcal{L}_{\text{total}}$
17: **end for**

---

## A.2 TRAINING DYNAMICS

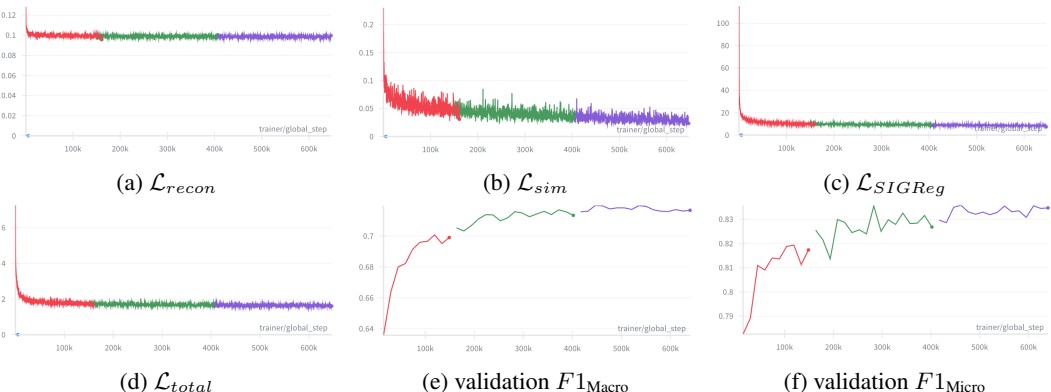

(a) $\mathcal{L}_{recon}$  (b) $\mathcal{L}_{sim}$  (c) $\mathcal{L}_{SIGReg}$

(d) $\mathcal{L}_{total}$  (e) validation $F1_{\text{Macro}}$  (f) validation $F1_{\text{Micro}}$

Figure 4: **Training dynamics of scJEPA.** (a–c) Individual loss components: reconstruction, similarity, and SIGReg. (d) Total loss. (e–f) cell type classification performance on the validation set. Different colors denote independent runs with varying hyperparameters.

Figure 4 illustrates the training dynamics of scJEPA. The total loss converges smoothly within the first 100k steps, demonstrating stable optimization. Among the individual components, the reconstruction loss $\mathcal{L}_{\text{recon}}$ stabilizes around 0.1, while the similarity loss $\mathcal{L}_{\text{sim}}$ exhibits steady decrease throughout training, indicating that local view embeddings progressively align with their corresponding global views. Notably, the SIGReg loss $\mathcal{L}_{\text{SIGReg}}$ drops rapidly in early training before plateauing, suggesting that the model quickly learns to produce well-distributed embeddings that avoid collapse—a key advantage of our principled regularization approach over heuristic methods. The downstream metrics show consistent improvement across runs, with $F1_{\text{Macro}}$ reaching approximately 0.73 and $F1_{\text{Micro}}$ exceeding 0.85 on validation data, confirming that the learned representations capture biologically meaningful structure.

