# OpenReview forum: "scJEPA: Dual-Space Self-Supervised Learning for Single-Cell Representations"
_ICLR.cc/2026/Workshop/LMRL — ICLR 2026 Workshop LMRL Submission_

### Official Review · Reviewer_ARon · 2026-02-22

**Rating:** 6
**Confidence:** 3

**Review:**

**Summary**

The authors introduce a framework that couples denoising loss with a invariance objective which is similar idea observed in VAE styles objective and targets them for single cell data. The authors have provided ample experimental results and have shown the importance of the individual components in their objective. However, implementation details of SIGRef is underspecified and clarifications in the claims provided by the author needs improvement.

**Strengths/Pros:**

- The paper is well written and Figure 1 clearly illustrates the pipeline and gradient flow
- The framework is pre trained and evaluated on multiple datasets and ablations showing out of distribution generalization of the framework.

**Weaknesses/Cons**

- The authors does not provide information on how SIGReg is implemented considering SIGReg uses the epps pulley statistics with an intergral function. The authors haven't provided information of how many directions they have sampled or how many samples were done per run and how they achieve numerical stability. Additionally specifics of the classifier, fine tuning parameters and data imbalance handling should also be provided for reproducibility.

- The final loss objective shows two different weighing schemes for  similarity vs SIGRef and reconstruction. The authors do not specify how these weights are obtained. are the weights learned on a smaller subset? or were other hyperparameter balancing techniques used?

- The authors claim a JEPA style latent prediction but the methodology and loss definition resonated more with invariance learning methods.

---

### Official Review · Reviewer_vgH5 · 2026-02-24
**A Principled Dual-Space SSL Framework for Single-Cell Data with Strong Empirical Gains but Limited Methodological Novelty**

**Rating:** 8
**Confidence:** 3

**Review:**

A theoretically grounded dual-space SSL framework (scJEPA) that integrates denoising and latent prediction, achieving strong empirical performance in single-cell representation learning.

### Pros
- Clear dual-space design separating information preservation (reconstruction) and selection (latent prediction).
- Principled collapse prevention via full distributional regularization (SIGReg).
- Consistent gains across zero-shot, OOD, integration, and spatial transfer tasks.

### Cons
- Limited methodological novelty; mainly an integration of existing components.
- Lacks comparison with recent large-scale transformer-based single-cell foundation models.
- No empirical isolation of SIGReg’s advantage over simpler regularization baselines.

---

### Meta-Review · Area_Chair_SRWn · 2026-02-25

**Recommendation:** Accept (Poster)
**Confidence:** 4

**Metareview:**

Accept

---

### Decision · Program_Chairs · 2026-03-09

Accept (Poster)